# Simultaneous perception of prosthetic and natural vision in AMD patients

D. Palanker [1✉], Y. Le Mer[2], S. Mohand-Said[3] & J. A. Sahel [2,3,4,5]

Loss of photoreceptors in atrophic age-related macular degeneration (AMD) results in severe visual impairment. Since the low-resolution peripheral vision is retained in such conditions, restoration of central vision should not jeopardize the surrounding healthy retina and allow for simultaneous use of the natural and prosthetic sight. This interim report, prespecified in the study protocol, presents the first clinical results with a photovoltaic substitute of the photoreceptors providing simultaneous use of the central prosthetic and peripheral natural vision in atrophic AMD. In this open-label single group feasibility trial (NCT03333954, recruitment completed), five patients with geographic atrophy have been implanted with a wireless 2 x 2 mm-wide 30 µm-thick device, having 378 pixels of 100 µm in size. All 5 patients achieved the primary outcome of the study by demonstrating the prosthetic visual perception in the former scotoma. The four patients with a subretinal placement of the chip demonstrated the secondary outcome: Landolt acuity of 1.17 ± 0.13 pixels, corresponding to the Snellen range of 20/460–20/565. With electronic magnification of up to a factor of 8, patients demonstrated prosthetic acuity in the range of 20/63–20/98. Under room lighting conditions, patients could simultaneously use prosthetic central vision and their remaining peripheral vision in the implanted eye and in the fellow eye.

[1] Department of Ophthalmology and Hansen Experimental Physics Laboratory, Stanford University, Stanford, CA, USA. [2] Department of Ophthalmology, Fondation Ophtalmologique A. de Rothschild, Paris, France. [3] Clinical Investigation Center INSERM-DGOS 1423, Quinze-Vingts National Eye Hospital, Paris, France. [4] Department of Ophthalmology, University of Pittsburgh School of Medicine, Pittsburgh, PA, USA. [5] Sorbonne Université, INSERM, CNRS, Institut de la Vision, Paris, France. ✉email: palanker@stanford.edu

Age-related macular degeneration (AMD) is a leading cause of irreversible vision loss[1], with its prevalence dramatically increasing in the aging population: from 1.5% in the US residents above 40 years to more than 15% in the subjects older than 80[2]. Currently, there is no efficient therapy for preventing the AMD progression, except for suppression of the neovascularization[3], although research in this field continues[4]. The atrophic form of AMD (also known as geographic atrophy, GA) results in a gradual loss of photoreceptors in the central macula, which is responsible for high-resolution vision, and severely impairs reading and face recognition. Low-resolution peripheral vision is retained in this condition, enabling orientation and the use of eccentric fixation for visual discrimination at reduced acuity. Therefore, the goal of any treatment strategy should be to restore functional central vision without jeopardizing the surrounding retina and allowing for their simultaneous use.

While photoreceptors gradually disappear in GA, the inner retinal cells survive to a large extent[5]. To restore sight in the scotoma, we replace the lost photoreceptors with photovoltaic pixels in the subretinal implant, which convert light into electric current to selectively stimulate the secondary neurons in the retina[6]. These electronic substitutes of photoreceptors replace the two main functions of the natural photoreceptors: (a) the light-to-current conversion, corresponding to the function of the outer segment, and (b) transfer of the visual information to secondary neurons by their polarization in extracellular electric field, substituting the function of the synapse.

To avoid irreversible electrochemical reactions at the electrode–electrolyte interface, stimulation current is pulsed and charge-balanced. On the other hand, to provide steady visual percepts under pulsatile illumination, repetition rate should exceed the frequency of flicker fusion. In preclinical studies, we demonstrated that selective stimulation of bipolar cells without direct activation of the downstream neurons results in preservation of multiple features of the natural retinal signal processing, including flicker fusion, adaptation to static images[6], ON and OFF responses with antagonistic center-surround[7], and non-linear summation of subunits in the retinal ganglion cells' (RGC) receptive fields[6]. We have also shown that grating visual acuity (VA) matches the pixel pitch with 75 and 55 μm pixels[6,8].

The first generation of the human-grade photovoltaic subretinal prosthesis PRIMA (Pixium Vision SA., Paris, France) is 2 mm in width (~7° of the visual angle in a human eye), 30 μm in thickness, containing 378 hexagonal pixels of 100 μm in width.

Images captured by the camera are processed and projected onto the retina from video glasses using intensified light (Fig. 1). To avoid photophobic and phototoxic effects of bright illumination, we use near-infrared (NIR, 880 nm) wavelength[9]. Photovoltaic pixels in the implant directly convert the projected pulsed light into local electric current flowing through the retina between the active and return electrodes[6,10].

## Results

Five patients with GA were implanted in Paris during 2017–2018 (NCT03333954). In four of them, the implant was placed in the subretinal space, but in one it ended up inside the choroid due to patient's accidental movement during surgery. In one of the four patients, the implant accidentally shifted by about 2 mm from the central position after the fluid-air exchange since the patient did not keep the head in a prone position post implantation. Due to wireless nature of the implant, surgical procedure was relatively short—about 2 h[11]. As shown in Table 1, residual natural acuity in the operated eye did not decrease in any of the subjects. Interestingly, in some patients, acuity improved compared to baseline, which could be attributed to either a neurotrophic benefit of subretinal surgery[12] or of electrical stimulation[13] or just improvement with eccentric fixation after training.

The primary endpoint of the study—prosthetic light perception measured in the visual field test (Octopus 900; Haag-Streit, Koniz, Switzerland), demonstrated that visual perception was elicited by the PRIMA implant in all subjects, as reported earlier for the time period of 6–12 months[11]. During the 18–24 months follow-up period, sensitivity improved in all subjects, as shown in Table 1, except for patient #3, who passed away due to unrelated cause before the second phase of the trial. In the first phase of the trial[11], prosthetic vision was assessed independently from the remaining natural vision. For this purpose, opaque virtual reality glasses (VR, PRIMA-1) have been used. The projected images covered a horizontal field of 5.1 mm (17.5° on the retina), with approximate resolution of 10.5 μm. Maximum peak retinal irradiance was 3 mW/mm², well within the thermal safety limits for chronic use of near-infrared light[14]. Brightness of the percept was controlled by pulse duration, between 0.7 and 9.8 ms, in 0.7 ms increments.

The four patients with subretinal implant placement demonstrated monochromatic (white-yellowish "sun-color") form perception, with flicker fusion above 30 Hz. In three patients with

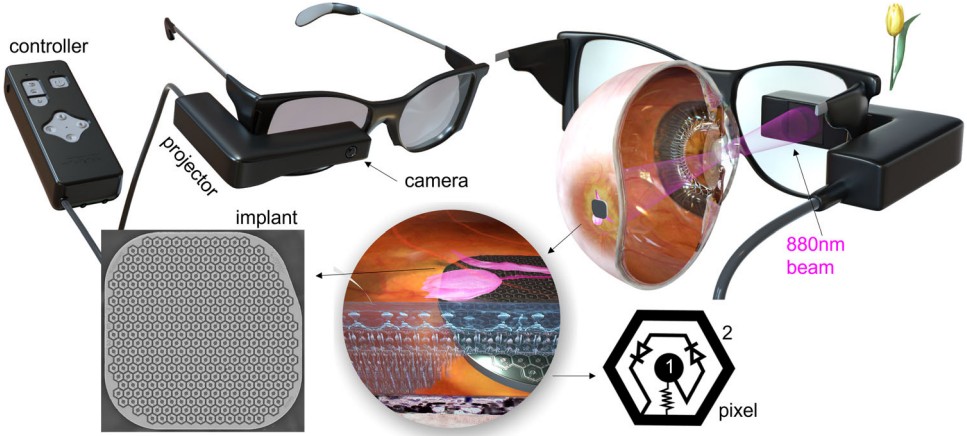

**Fig. 1 Diagram of the PRIMA system.** Top row: Artistic rendering of the augmented reality glasses with a projector and a camera. The 880 nm beam projects the video stream onto the retina. Bottom row: PRIMA implant with a hexagonal array of 100 μm pixels. Implant is placed under the degenerate retina without damaging the peripheral healthy retina. Pixels are composed of two photodiodes connected in series between the active (1) and a circumferential return (2) electrode.

**Table 1 Residual natural vision, anatomical and functional outcomes with the implant.**

| Test/Patient | 1 | 2 | 3 | 4 | 5 |
|---|---|---|---|---|---|
| *Patients' data* | | | | | |
| Age at baseline | 83 | 66 | 82 | 69 | 74 |
| Years of VA < 20/400 in implanted eye | 6 | 2 | 3 | 5 | 10 |
| *Residual peripheral vision* | | | | | |
| Pre-op natural letter acuity in the study eye | 20/400 | 20/800 | 20/1000 | 20/500 | 20/500 |
| Postop natural letter acuity in the study eye at 12 months and at 24 months | 20/320; 20/160 | 20/800; 20/200 | 20/800* | 20/400; 20/500 | 20/400; 20/550 |
| Pre-op letter acuity in the fellow eye | 20/100 | 20/50 | 20/125 | 20/400 | 20/100 |
| Postop letter acuity in the fellow eye at 12 months and at 24 months | 20/160; 20/160 | 20/50; 20/50 | 20/200* | 20/400; 20/640 | 20/125; 20/125 |
| *Implant location* | | | | | |
| Implant location in the macula | Intra-choroidal | Central subretinal | Central subretinal | Off-center subretinal | Central subretinal |
| *Stimulation threshold and sensitivity* | | | | | |
| Perceptual threshold with PRIMA-1 glasses, ms | 2.1 | 0.8 | 0.7 | 1.0 | 0.8 |
| Perceptual threshold with PRIMA-2 glasses, ms | 1.28 ± 0.84 | 0.75 ± 0.19 | * | 0.82 ± 0.29 | 0.70 ± 0.00 |
| Central perceptual threshold in OCTOPUS, dB at 6–12 and at 18–24 months | 0.9; 2.5 | 1.3; 1.9 | 3.1* | 0.4; 2.9 | 1.3; 10.1 |
| *Prosthetic visual acuity* | | | | | |
| PRIMA-1 (VR), 12 months; no magnification Min. Landolt C gap, pix | Light perception | 20/550; logMAR 1.44 1.3 pix | 20/500; logMAR 1.40 1.2 pix | 20/800; logMAR 1.60 1.9 pix | 20/460; logMAR 1.37 1.1 pix |
| PRIMA-2 (AR), 18–24 months, no magnification Min. Landolt C gap, pix | Light perception | 20/564; logMAR 1.45 1.34 pix | * | 20/438; logMAR 1.34 1.04 pix | ** |
| PRIMA-2 (AR), Landolt VA with preferred magnification, 18–24 months | Light perception | 20/98; logMAR 0.69 | * | 20/71; logMAR 0.55 | 20/63; logMAR 0.50 |
| Natural Landolt VA in the study eye, 18–24 months | 20/182; logMAR 0.96 | 20/246; logMAR 1.09 | * | 20/428; logMAR 1.33 | 20/332; logMAR 1.22 |
| LogMAR gain due to PRIMA 18–24 months | 0 | 0.4 | * | 0.78 | 0.72 |
| *Background lighting* | | | | | |
| Background light threshold [cd/m²] | NA, no shape perception | >256 | * | >256 | 64 |
| Attenuation for bright room lighting | NA | Clear | * | Clear | 65% |
| *Simultaneous perception of prosthetic and natural vision* | | | | | |
| Bar orientation, % correct Monocular | NA, | 100 | * | 100 | 96 |
| Binocular | No shape perception | 100 | | 100 | 92 |

*Patient 3 passed away before the second phase of the trial.
**Patient 5 was not available for this measurement because of the COVID restrictions.

central location of the subretinal implant, acuity closely matched the pixel size: 20/460, 20/500, and 20/550 (1.1, 1.2, and 1.3 pixels). Patient with the off-center implant demonstrated lower acuity: 20/800 (1.9 pixels)[11]. Patient #1 with the intra-choroidal implant had blurry prosthetic vision, with no discernable acuity.

In the second phase of the study, starting at 18–24 months post-op, we introduced augmented reality glasses (AR, PRIMA-2), which allow unobstructed natural vision by the fellow eye and by the peripheral field of the operated eye, simultaneously with prosthetic central vision in the treated eye (Fig. 2a). The images, projected through the glasses adapted to patient's refraction, covered a horizontal field of 5.3 mm (18.5°) on the retina, with a resolution of 6.7 μm, as illustrated in Fig. 2b. This design provided improved beam homogeneity and easier alignment, compared to VR glasses (PRIMA-1). The maximum retinal irradiance was increased to 3.5 mW/mm², with the same range of pulse durations as with the VR glasses. This system allows the use of electronic magnification (×1, ×2, ×4, and ×8) between the camera and the image projection onto the implant. As shown in Table 1, perceptual thresholds 18–24 months after the implantation, measured with PRIMA-2 glasses, were slightly lower than the

thresholds measured during the first 6 months—in the first phase of the trial[11].

Prosthetic visual acuity with PRIMA-2 glasses was measured using Landolt C optotypes. To mimic the crowding effect of the letter charts, the Landolt rings were surrounded by a square frame (Fig. 3a). At each trial, subjects reported the font orientation (up, down, left, or right), and its size was then adjusted, depending on the response. The visual acuity was determined using the Freiburg Visual Acuity Test (FrACT) software[15,16]. For a stable perception under pulsatile illumination, 30 Hz repetition rate was applied. In the first set of the tests, computer-generated Landolt optotypes were projected into the eye directly from the AR glasses without using a camera. As shown in Table 1, patients #2 and #5 demonstrated prosthetic acuity at the level similar to that observed with VR glasses in the first phase of the trial (20/500, 20/460), but patient #4 improved compared to the earlier result—from 20/800 to 20/438. This is potentially due to easier alignment of the display to the off-center location of the implant with improved glasses. The average acuity in the four patients with the subretinal implant placement was 1.17 ± 0.13 pixels at the latest measurement, corresponding to logMAR 1.39, or 20/500 on a Snellen scale.

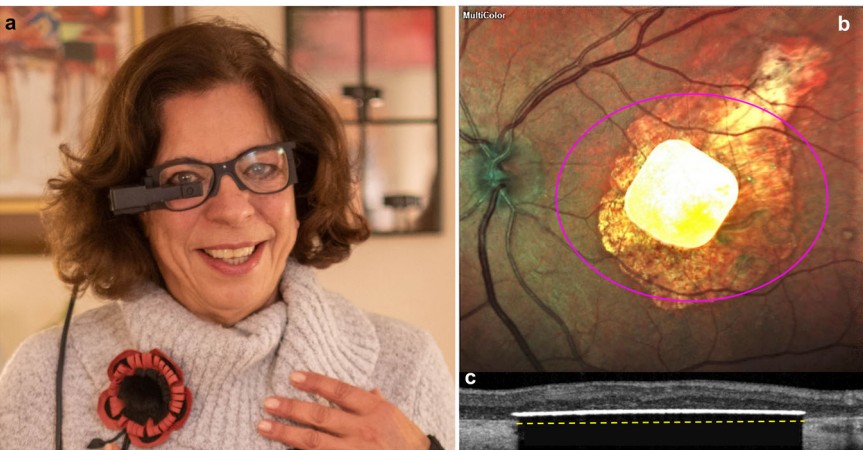

**Fig. 2 PRIMA system in practice. a** PRIMA-2 glasses on a person. **b** Fundus photo of a patient with the PRIMA implant inside the geographic atrophy area. Magenta oval illustrates the size of the beam (5.3 × 4.3 mm) projected onto the retina. **c** OCT image demonstrates the implant in subretinal space 6 months post-op. Yellow dash line depicts the approximate position of the back side of the implant resting on the Bruch's membrane.

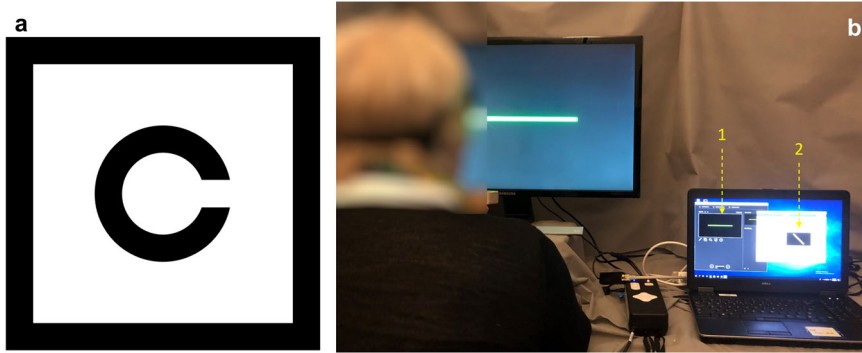

**Fig. 3 Assessment of prosthetic vision. a** Landolt C in the frame mimicking the crowding effect. **b** Testing setup with a patient sitting 40 cm from the screen. Horizontal green bar shown on a large display (1) can be seen with the remaining natural vision, while the diagonal bar (2) is presented only on the NIR display inside the glasses.

In the second set of the acuity tests, Landolt C optotypes were displayed at 40 cm distance from the subject, so patients used the camera and were allowed to apply their preferred electronic magnification (1, 2, 4, or 8). To ensure that prosthetic acuity is measured rather than the residual natural vision, in these tests the fellow eye was covered. In addition, contrast of the electronic image was inverted from the original black optotypes on a white background to white optotypes on a black background (white patterns stimulate the retina), and patients were asked about the color of the percept. With magnification, all three participants of the second trial could recognize optotypes equating to 20/98, 20/71, and 20/63 acuity, respectively. As shown in Table 1, these values significantly exceeded their residual natural acuity in the treated eye, and for patients 4 and 5, even in the (better) fellow eye.

Video S1 illustrates a patient's letter recognition using prosthetic vision, with ×4 magnification and a contrast reversal. Video S2 illustrates a word reading test with ×4 magnification and a control experiment (Video S3), where patient is attempting to read the same word without the PRIMA glasses. It is important to emphasize that these videos are for illustration purposes only. Quantitative measurements of the prosthetic acuity were conducted with Landolt C optotypes.

To evaluate the effect of background light on prosthetic vision when the transparent AR glasses are used, Landolt C optotypes have been presented on the glasses display directly, without using the camera, while intensity of the background visible light was varied. In this experiment, subjects with both eyes open were placed 40 cm in front of a wide LCD screen, where a homogeneous white illumination at 16 levels (ranging from 1.4 to 256 cd/m²) was presented. Prosthetic patterns were presented at maximum brightness: 3.5 mW/mm² of NIR irradiance with 9.8 ms pulse duration. As shown in Table 1, subjects 2 and 4 did not have a problem seeing the Landolt C in front of the screen even with the highest background luminance (256 cd/m²). Subject 5 had difficulties with luminance above 64 cd/m², and therefore was provided later with a shaded lens (65% attenuation of white light) to allow using the device in a bright office environment.

It is important to note that patients could simultaneously use prosthetic and residual natural vision from both, the study eye, and the fellow eye. For example, in a setup shown in Fig. 3b, green bars of various orientations were presented on a large screen for natural vision and another set of bars was simultaneously presented just on the NIR display inside the glasses. The patients were asked about both orientations and colors, as illustrated in the Video S4 for binocular vision and in the Video S5 for monocular vision. In both cases, the bars were perceived simultaneously at correct orientations, as summarized in the Table 1.

## Discussion

This study was limited primarily to the clinical evaluation of prosthetic vision. Future testing will be expanded to include a

home use, which will increase the active time and also help assess the functional benefits of the device, considering the residual vision available in the fellow eye. Future studies will also assess a practical value of the electronic magnification for the central vision in the PRIMA system.

In summary, this trial confirmed the safety and stability of the PRIMA implant over 18–24 months follow-up in five patients with geographic atrophy. Prosthetic central vision in the former scotoma represents monochromatic form perception matching the presented patterns and, most importantly, is perceived in conjunction with the residual peripheral vision, thus enabling natural orientation and central discrimination. Spatial resolution was, on average, 1.2 pixels of the implant, corresponding to acuity of about 20/500, and using electronic magnification, all patients with subretinal implant demonstrated acuity exceeding 20/100. Recent advancements in the photovoltaic pixel design, which enable five times the smaller pixels[17,18] may improve acuity of the future PRIMA devices up to 20/100, and with electronic zoom—potentially up to 20/20. Further improvements with video glasses may widen the visual field, while the advanced image processing and stimulation protocol may help enhance the dynamic range and contrast sensitivity, promising even more functional restoration of sight for numerous patients suffering from atrophic macular degeneration.

## Methods

**Patients**. The aim of this study (NCT03333954) was to test functionality of the PRIMA system in 5 patients with atrophic AMD. The study adhered to the Declaration of Helsinki and received the ethics committee approval from the Comité de Protection des Personnes Ile de France II and approval by the Agence Nationale de Sécurité du Médicament et des Produits de Santé in France (ASNM). Study participants were above 60 years of age and had advanced atrophic AMD with an atrophic zone of at least three optic disc diameters and best corrected visual acuity of ≤20/400 in the worse-seeing study eye; no foveal light perception (absolute scotoma) but visual perception in the periphery, with preferred retinal locus determined by micro-perimetry; absence of photoreceptors and presence of the inner retina in the atrophic area as confirmed by optical coherence tomography (OCT); absence of choroidal neovascularization verified by retinal angiography. Patient #2 had glaucoma, while all other ocular and general pathologies that could contribute to the low visual acuity were excluded. All patients, except for #1, underwent visual rehabilitation for improving eccentric fixation before they were recruited for the trial. Patients provided written informed consent to participate in the study and to publish clinical information and videos in anonymized form. Person shown in Fig. 2a is a healthy volunteer who provided a written informed consent to publish her photo in a scientific journal. Patients recruitment was completed in 2018.

Implantation took place at the Foundation A Rothschild Hospital (Paris, France). The patients' rehabilitation and visual function assessment were carried out at the Clinical Investigation Center of Quinze-Vingts National Eye Hospital (Paris, France). In addition to the primary endpoint of the study, the prosthetic light perception measured in the visual field test, a secondary endpoint was added in 2019: the visual acuity measured by Landolt C optotypes, as well as exploratory studies of the visual function, including the simultaneous use of the natural and prosthetic vision. These modifications were approved by ASNM and by the same ethics committee as the original protocol. Results in this paper represent an interim report prespecified in the study protocol.

Lab tests were conducted, on average, once a week before COVID, but during the pandemic the tests frequency significantly decreased due to restrictions on patients' travel. As these limitations have been relaxed, the frequency of the tests is being increased again.

**Assessment of prosthetic vision**. Visual acuity was assessed using a computer-generated Landolt C in four different orientations (gap at the top, bottom, right or left), so that a random response corresponds to 25% accuracy. The threshold optotype size was defined as the proper symbol recognition with at least 62.5% accuracy. To minimize the number of presentations, the study was conducted using the method of the Freiburg Acuity and Contrast Test (FrACT)[16], which was shown to yield equivalent VA when measured with Landolt C optotypes to that obtained with ETDRS charts[19]. In this protocol, a single Landolt C is presented in a fixed central position on a display, and the best Parameter Estimation by Sequential Testing (best PEST) procedure[20] was used to estimate the VA. Since it is an adaptive test, the number of times that each optotype is presented varies depending on the patient responses. The test was performed three times on three different days. At each day, 24 trials were performed twice, and the mean of these two runs

was considered the daily result. The final result was defined as the median of the three daily measurements. To mimic the crowding effect of the letters in vision charts, Landolt C was presented with a frame around it (Fig. 3a).

**Background illumination**. To provide a uniform and controllable background illumination, a 71 cm-wide Samsung U28E590D LCD screen has been used. Subjects were placed 40 cm in front of a screen, where a homogeneous white illumination at 16 levels was presented: 256, 181, 128, 90.5, 64, 45.3, 32, 22.6, 16, 11.3, 8, 5.7, 4, 2.8, 2, 1.4 cd/m², while room lights were turned off.

The best-PEST (Parameter Estimation by Sequential Testing) method[20] was used to determine the maximum luminance which still allowed identifying the prosthetic patterns (Landolt C optotypes with four different orientations) with accuracy exceeding 62.5% using the following parameters: 20 iterations, 0.25 false positive rate, 0 false negative rate, and sigmoid slope of 0.5. The last value of the best-PEST was used as the resulting threshold luminance. Data for the three patients with a subretinal implant are presented, but not for subject #1 since this subject with intra-choroidal chip placement could not recognize Landolt C. Prosthetic patterns were presented at maximum brightness: 3.5 mW/mm² of NIR irradiance with 9.8 ms pulse duration and 30 Hz repetition rate. The patterns were presented for up to 30 s, with 10 s break between the stimuli.

**Simultaneous perception of prosthetic and natural vision**. A green bar was displayed on an LCD screen placed at 40 cm in front of the subject in one of the four possible orientations (vertical, horizontal, 45° diagonal from the upper left, 45° diagonal from the lower left). Simultaneously another bar is also projected in Artificial Pattern Mode (APM) on the PRIMA Glasses in one of the four orientations (Fig. 3b). The width of the bars corresponds to 0.4 mm on the retina (four implant pixels). The bar orientation on LCD screen is random, but 50% of the times orientation of the bar displayed on the glasses matches the bar orientation on LCD. The subject is expected to see the bar displayed on the LCD screen with the natural peripheral vision and report it as green, and the bar projected by the PRIMA Glasses with prosthetic vision, perceived as white. At each repetition, the subject is asked about orientation of each bar individually.

A total of 48 bar pairs were presented (24 presentations with the fellow eye open and 24 presentation with the fellow eye closed), each for a duration of up to 20 s. The subjects were not allowed to move their head—only the eyes.

**Reporting summary**. Further information on research design is available in the Nature Research Reporting Summary linked to this article.

## Data availability

All the data supporting the findings of this study are available within the paper and its supplementary information files. The visual acuity data analysis software can be downloaded from the publicly available database[21]. Protocol for the studies of simultaneous perception of prosthetic and natural vision can be provided upon a reasonable request. Pixium Vision is responsible for approval of the clinical study protocol and for reporting the results to the regulatory authorities in Europe and in US. As such, it was informed about the study on a continuous basis and performed its own data analysis. The paper was written by its authors based on their data analysis, and they are responsible for its content.

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

## Acknowledgements
The authors thank the patients who participated in the study; the Pixium Vision team who designed, fabricated, and tested the PRIMA system; the Scientific and Medical Advisory Board of Pixium Vision for its guidance on the clinical trial design; and all the scientific, research and development, medical, and clinical research staff who continue the patient care, rehabilitation, and evaluation. Studies were supported by: Pixium Vision SA; the Sight Again project (via Structural R&D Projects for Competitiveness and Investment for the Future funding managed by BpiFrance) and the Clinical Investigation Center at the Quinze-Vingts National Hospital, which is supported in part by the Inserm-DGOS, France and by LabEx LIFESENSES (ANR-10-LABX-65) and IHU FOReSIGHT (ANR-18-IAHU-01) grants. D.P. is supported in part by the National Institutes of Health (R01-EY027786). J.A.S. is supported in part by the NIH CORE grant P30 EY08098, and by unrestricted grant from Research to Prevent Blindness, New York.

## Author contributions
Y.L.M., S.M.S., J.A.S., and D.P. conceptualized the study, Y.L.M. screened the patients, performed implantations and post-op imaging. S.M.S. guided the rehabilitation and assessment of prosthetic vision. D.P., Y.L.M., J.A.S., and S.M.S. analyzed the data and wrote the manuscript.

## Competing interests
D.P.: Consultant, and Patent Royalties with Pixium Vision, Y.L.M.: Consultant with Pixium Vision, J.A.S.: Equity owner at Pixium Vision, S.M.S.: Consultant with Pixium Vision.

## Additional information

**Peer review information** *Nature Communications* thanks James Weiland, Maureen Maguire and the other anonymous reviewer(s) for their contribution to the peer review this work. Peer reviewer reports are available.

