## [Peer Review File · Nature Communications]

Reviewers' Comments:

Reviewer #2:

Remarks to the Author:

I am a medical statistician and have been working in ophthalmology for 12 years and have worked in the area of AMD for 4 years.

I found the paper easy to read, it is clear and well-written. The videos were fun but I am not sure what they added.

There is no mention in the background section of the current therapies of AMD, the overall success and the need for alternative treatments. Maybe mention ageing populations, independence.... This is going to an eye journal but may be read by people interested in the tech.

In Table 1 could you include some information on the patients? Age, time since diagnosis of AMD, any other eye conditions, previous treatments. Vision in the contra-lateral eye, and maybe compare change in the treated eye with change in the untreated eye.

It may be there but I don't think the acceptability of this treatment by the patients is mentioned. Was there any qualitative work done with these patients?

In the discussion maybe add a couple of sentence about future work.

Reviewer #4:

Remarks to the Author:

The authors provide results from 5 patients who had a prosthetic device meant to restore central vision to patients with the advanced late form of age-related macular degeneration (AMD), geographic atrophy. Results through 1 year have been reported previously. This report extends the data on vision in the study eye using prosthetic vision through 2 years and adds an assessment of visual function while using both prosthetic vision and (peripheral field vision in the study eye and contralateral eye vision) – more usual conditions for viewing. An improvement in performance was achieved by improving the projection of the infrared light onto the device using augmented reality glasses rather virtual reality glasses. These findings provide proof-of-concept evidence for the approach under study.

1. Limitations of this report: One of the 5 patient had the device implanted in an unintended position and does not contribute meaningful data and one of the patients died, leaving only 3 patients with evaluable data at 2 years.

2. The authors should mention and support with references whether comparable visual acuity measurements are obtained with letter optotypes and Landolt C optotypes. Were patients refracted at each visit and tested with updated prescriptions?

3. Lines 99- 101. The assertion that there was improvement in performance by improving the projection of the infrared light onto the device using augmented reality glasses rather virtual reality glasses may not be true. The change from 20/800 to 20/438 may not represent a significant improvement in acuity because even with standard refraction and visual acuity testing regimens, test-retest variability can result in a 2 line (20/800 to 20/500) improvement in low vision patients.

More minor comments:

4. Line 138. Data presented in this report was through 18-24 months, not 24-30.

5. Table 1. This table has the key results but is difficult to follow. The following suggestions may improve the presentation of the data.

- Using slightly different labels for the same feature increases complexity. I assume "eccentric natural letter acuity" means the same as "natural letter acuity" and that "PRIMA" in the last line of the Prosthetic visual acuity section is "PRIMA-2 (AR)".

- Measures of the same quantity have different numbers of decimal places – are the greater number of decimal places warranted by the precision of the measurements? See perceptual thresholds, Landolt D gaps, and LogMAR acuities.

- Does pix provide additional information to the Snellen equivalent and LogMAR values for acuity?

6. The data in Table 1 for features measured preoperatively and through 12 months have been published previously in Ophthalmology, as have some of the figures and patient images. The editorial team should advise on whether permissions are required.

Reviewer #5:

Remarks to the Author:

The authors report on visual acuity measures and other tasks using the Prima subretinal implant. The new component of this work is improved glasses (version AR), which allow the implant patient to use both the PRIMA and their remaining peripheral vision. The authors report maintained visual acuity (VA) with the PRIMA at up to 24 months (compared to a prior publication) and in one case, significantly improved VA. Adding a zoom feature on the glasses allowed patients to achieve scores on VA tests equivalent to someone with VA of 20/63-20/98.

The current manuscript reports important findings but has major flaws and also some statements that omit important facts.

The important findings are

1) maintained VA at 24 months. This is important for two reasons. First, the retina is tolerating well the implant. Second, the implant, which has a unique protective coating, is still functioning. It is not clear how often the implant is activated. If this is only occasionally in clinic, then the second result is less surprising. The authors should provide details on the frequency of implant use.

2) Simultaneous integration of natural and prosthetic vision. This is key to the success of prosthetic approach to AMD. Unfortunately, the only data shown is a video of a single patient. The methods describe presentation of 48 "bar pairs", but the results of these repeated tests are not provided. The authors are encouraged to design a rigorous test and report the results from all study participants.

The inclusion of a zoom feature is also an important result, but at the same time, the experiment used to determine the benefit of this feature was flawed. With a user controlled magnification, the patients used the Prima to recognize letters of a size equating VA of 20/62-20/91. The authors failed to have the user try this same task with a simple magnifying glass with the study eye. The correct comparison is magnified prosthetic vision vs. magnified natural vision.

The methods describe VA estimate using a Landolt C, as part of the BALM set of visual acuity tests. These tests are validated and well described in the methods and in prior literature. In contrast, the second visual acuity test, using letter reading, has no description in the methods. The results section provides a brief description of how the test was conducted, but there is no information on % correct, how much time was allotted for the user to determine the letter, and the result repeatability. Thus, the results for letter reading should be considered anecdotal, unless the authors can provide more detail here on the methods used to arrive at these VA numbers.

Related to the claim of letter reading VA, the authors should revise lines 111-115. The results are over-stated. It is arguable if acuity (the spatial resolution of retinal activation) can really be changed by magnification. If this definition for acuity is accepted, then conceivably a 10,000X magnification could achieve enormous gains in VA. Is it super-human acuity if someone views images from a microscope using VR glasses? Not really. A more accurate statement for this result is "user controlled magnification allowed patients to recognize letters equating to 20/63 if viewed with natural vision". It is still a good result and user's controlling the magnification will allow them to select their best settings. But claiming improved acuity through magnification is dubious.

We would like to thank the editor and the reviewers for careful reading of the manuscript and for their comments and questions. We addressed them below and made corresponding changes in the manuscript.

Editor: please revise the manuscript text to address our editorial requests regarding reporting of the clinical trial as outlined in the original decision letter (dated May 14th). In particular, please signpost the primary, secondary and additional exploratory outcomes in the main text, signpost any deviations from the study protocol/modifications in the outcomes and confirm that any modifications were approved by the institutional review board in the methods section, and indicate that this is an interim report prespecified in the study protocol in the abstract and the methods section. We also request a CONSORT checklist for pilot and feasibility studies to facilitate editorial and reviewer assessment.

We now included in the main text (page 3) a description of the primary endpoint of the study - the prosthetic light perception measured in the visual field test. We also stated in the Methods (Patients section) that a secondary endpoint, the visual acuity measured by Landolt C optotypes, was added in 2019, and approved by ASNM - Agence Nationale de Sécurité du Médicament et des Produits de Santé in France (equivalent of the FDA) and by the same ethics committee as the original protocol (Comité de Protection des Personnes Ile de France II). We also stated that the original protocol included exploratory studies of the simultaneous use of the natural and prosthetic vision. We indicated that results in this paper represent an interim report prespecified in the study protocol. The CONSORT checklist is attached as a separate document.

Reviewer #2:

I am a medical statistician and have been working in ophthalmology for 12 years and have worked in the area of AMD for 4 years. I found the paper easy to read, it is clear and well-written. The videos were fun but I am not sure what they added.

There is no mention in the background section of the current therapies of AMD, the overall success and the need for alternative treatments. Maybe mention ageing populations, independence.... This is going to an eye journal but may be read by people interested in the tech.

We now added a sentence (2nd) and references about the current therapies and development of alternative treatments for AMD. We now also explicitly mentioned the aging population. Restriction on the text length and the number of references precludes us from going into more details.

In Table 1 could you include some information on the patients? Age, time since diagnosis of AMD, any other eye conditions, previous treatments. Vision in the contra-lateral eye, and maybe compare change in the treated eye with change in the untreated eye.

Vision in the treated and in the contralateral eye is already listed in the Table, pre-op, at 12 and 24 months post-op. We now added to the Table 1 the age and the time since VA dropped below 20/400. We also added to the Patients section in the Methods the info about visual rehabilitation prior to surgery and information on other eye conditions.

It may be there but I don't think the acceptability of this treatment by the patients is mentioned. Was there any qualitative work done with these patients?

In this feasibility trial, the device was mainly used in the lab testing. Therefore, no representative data for acceptability of the system in daily living is available. We plan to explore this in the next phase of the trial.

In the discussion maybe add a couple of sentence about future work.

We now added two sentences at the end of the Discussion section about future work: “Recent advancements in the photovoltaic pixel design, which enable the 5 times smaller pixels may improve acuity of the future PRIMA devices up to 20/100, and with electronic zoom – potentially up to 20/20. Further improvements with video glasses may widen the visual field, while advanced image processing and stimulation protocol may help enhance the dynamic range and contrast sensitivity, promising even more functional restoration of sight for numerous patients suffering from atrophic macular degeneration. “

Reviewer #4:

The authors provide results from 5 patients who had a prosthetic device meant to restore central vision to patients with the advanced late form of age-related macular degeneration (AMD), geographic atrophy. Results through 1 year have been reported previously. This report extends the data on vision in the study eye using prosthetic vision through 2 years and adds an assessment of visual function while using both prosthetic vision and (peripheral field vision in the study eye and contralateral eye vision) – more usual conditions for viewing. An improvement in performance was achieved by improving the projection of the infrared light onto the device using augmented reality glasses rather virtual reality glasses. These findings provide proof-of-concept evidence for the approach under study.

1. Limitations of this report: One of the 5 patient had the device implanted in an unintended position and does not contribute meaningful data and one of the patients died, leaving only 3 patients with evaluable data at 2 years.

2. The authors should mention and support with references whether comparable visual acuity measurements are obtained with letter optotypes and Landolt C optotypes.

We now added to the section Assessment of prosthetic vision in the Methods a statement and a reference [19] about equivalence of the visual acuity measurements using Freiburg Acuity and Contrast Test (FrACT) [16] with Landolt C optotypes to that measured with ETDRS letter charts. As shown in Figures 2B and 3 in ref [19] (image on the right), visual acuity measured by FrACT and by ETDRS charts coincide without a systematic bias and with a mean variation of 15% over a wide range of these measurements: from 0.01 to 0.3 in fractional acuity. This range covers the acuity span of our measurements: 0.025 – 0.3.

Were patients refracted at each visit and tested with updated prescriptions?

Yes, all the VA measurements were performed with properly corrected refraction.

3. Lines 99- 101. The assertion that there was improvement in performance by improving the projection of the infrared light onto the device using augmented reality glasses rather virtual reality glasses may not be true. The change from 20/800 to 20/438 may not represent a significant

improvement in acuity because even with standard refraction and visual acuity testing regimens, test-retest variability can result in a 2 line (20/800 to 20/500) improvement in low vision patients.

We report the acuity as measured by the Freiburg Visual Acuity Test with the Landolt C optotypes. We now removed the word “significantly” from this sentence.

Minor comments:

4. Line 138. Data presented in this report was through 18-24 months, not 24-30.

Corrected, thank you.

5. Table 1. This table has the key results but is difficult to follow. The following suggestions may improve the presentation of the data.

- Using slightly different labels for the same feature increases complexity. I assume “eccentric natural letter acuity” means the same as “natural letter acuity”

Yes. We removed the word “eccentric”.

- Measures of the same quantity have different numbers of decimal places – are the greater number of decimal places warranted by the precision of the measurements? See perceptual thresholds, Landolt D gaps, and LogMAR acuities.

Precision of the averaged data is limited to 2 digits after the decimal point throughout the Table. We omitted the second digit if it was 0, but we added it now for uniformity. With PRIMA-2, the perceptual threshold was measured more precisely than in the previous study with PRIMA-1 glasses, including the standard deviation and the second digit after the decimal point.

- Does pix provide additional information to the Snellen equivalent and LogMAR values for acuity?

Yes, resolution in pixels on the retina provides a direct comparison of the sampling limit (1 pix) to the actual acuity in each patient.

6. The data in Table 1 for features measured preoperatively and through 12 months have been published previously in *Ophthalmology*, as have some of the figures and patient images. The editorial team should advise on whether permissions are required.

Yes, Table 1 includes the longer follow-up now, but it also shows some of the 12 months data for comparison. Patient images in Figure 2A and OCT in 2B are new, and fundus photo in 2B has a different contrast, vertical size and is overlaid with a beam outline.

Reviewer #5:

The authors report on visual acuity measures and other tasks using the Prima subretinal implant. The new component of this work is improved glasses (version AR), which allow the implant patient to use both the PRIMA and their remaining peripheral vision. The authors report maintained visual acuity (VA) with the PRIMA at up to 24 months (compared to a prior publication) and in one case, significantly improved VA. Adding a zoom feature on the glasses allowed patients to achieve scores on VA tests equivalent to someone with VA of 20/63-20/98. The current manuscript reports important findings but has major flaws and also some statements that omit important facts. The important findings are

- 1) maintained VA at 24 months. This is important for two reasons. First, the retina is tolerating well the implant. Second, the implant, which has a unique protective coating, is still functioning. It is not clear how often the implant is activated. If this is only occasionally in clinic, then the second result is less surprising. The authors should provide details on the frequency of implant use.

Lab tests were conducted, on average, once a week before COVID, but during the pandemic the tests frequency significantly decreased due to restrictions on patients' travel. As these limitations have been relaxed, the frequency of the tests is being increased again. We added this information to the Patients section in Methods.

2) Simultaneous integration of natural and prosthetic vision. This is key to the success of prosthetic approach to AMD. Unfortunately, the only data shown is a video of a single patient. The methods describe presentation of 48 "bar pairs", but the results of these repeated tests are not provided. The authors are encouraged to design a rigorous test and report the results from all study participants.

We now added two lines to the Table 1, quantifying the bar orientation test for monocular and binocular prosthetic and natural vision.

3) The inclusion of a zoom feature is also an important result, but at the same time, the experiment used to determine the benefit of this feature was flawed. With a user controlled magnification, the patients used the Prima to recognize letters of a size equating VA of 20/62-20/91. The authors failed to have the user try this same task with a simple magnifying glass with the study eye. The correct comparison is magnified prosthetic vision vs. magnified natural vision.

This test was conducted only with an electronic magnifier and therefore was only applied to the prosthetic vision. If one would like to assess performance of the natural vision under similar magnification conditions, it should include a similar camera and projector, but at the visible wavelengths. Magnifying glass has very different working distance and a field of view, and hence does not necessarily provide an adequate comparison.

4) The methods describe VA estimate using a Landolt C, as part of the BALM set of visual acuity tests. These tests are validated and well described in the methods and in prior literature. In contrast, the second visual acuity test, using letter reading, has no description in the methods. The results section provides a brief description of how the test was conducted, but there is no information on % correct, how much time was allotted for the user to determine the letter, and the result repeatability. Thus, the results for letter reading should be consider anecdotal, unless the authors can provide more detail here on the methods used to arrive at these VA numbers.

Indeed, the only quantified results in terms of VA are with Landolt C, as listed in Table 1. Other tests are just illustrations of the patients' performance.

Related to the claim of letter reading VA, the authors should revise lines 111-115. The results are overstated. It is arguable if acuity (the spatial resolution of retinal activation) can really be changed by magnification. If this definition for acuity is accepted, then conceivably a 10,000X magnification could achieve enormous gains in VA. Is it super-human acuity if someone views images from a microscope using VR glasses? Not really. A more accurate statement for this result is "user controlled magnification allowed patients to recognize letters equating to 20/63 if viewed with natural vision". It is still a good result and user's controlling the magnification will allow them to select their best settings. But claiming improved acuity through magnification is dubious.

We revised this sentence: "With magnification, all three participants of the second trial could recognize letters equating to 20/98, 20/71 and 20/63 acuity, respectively."

Reviewers' Comments:

Reviewer #2:

Remarks to the Author:

Thank you for your careful replies to my questions and comments.

Reviewer #4:

Remarks to the Author:

The authors have responded sufficiently to the comments of the reviewers.

Reviewer #5:

Remarks to the Author:

The authors have clarified and improved the manuscript in this revision. There are several important results reported, as noted in my initial review. Several issues need to be addressed.

1. The authors have clarified the ETDRS letter reading tests "are just illustrations of patients' performance". Therefore, these should not be included in a results table with Landolt C VA. Including ETDRS results together with Landolt C VA results implies that the ETDRS results were obtained rigorously. The authors state that ETDRS results were anecdotal. The method sections does not include a description of ETDRS protocol and the result section does not report percent correct. I recommend the authors explicitly describe the limitations of ETDRS results and clearly separate ETDRS results from the Landolt C VA results by placing them in a different table.

2. The last paragraph should include a section on limitations of the study. Limitations include

a. The lack of take home testing limits the cumulative active time for PRIMA to a few hundred hours. Thus, claims of safety and stability for 24 months need to be qualified with a statement that the PRIMA was not used chronically, but rather implanted chronically and used occasionally.

b. The zoom test did not include a control condition for comparison. I agree with the authors' rebuttal that an electronically magnified display is the correct control. Such devices are readily available with commercial systems such as NuEyes and e-sight, and the AR glasses used for PRIMA can likely support this function. Is PRIMA an improvement over an electronically magnified visible light display, which does not require an implant?

c. An improvement in functional vision is not demonstrated in these patients, considering the level of vision available in the fellow eye. Therefore, at its current state, PRIMA will have limited impact, since patients will likely use the natural vision in their better eye for tasks that require high acuity vision. The author note several on-going improvements to address this issue.

We would like to thank the reviewers for carefully reading the manuscript, as well as for the questions and comments. We addressed them below and made corresponding changes in the manuscript.

Reviewer #5:

1. The authors have clarified the ETDRS letter reading tests “are just illustrations of patients’ performance”. Therefore, these should not be included in a results table with Landolt C VA. Including ETDRS results together with Landolt C VA results implies that the ETDRS results were obtained rigorously. The authors state that ETDRS results were anecdotal. The method sections does not include a description of ETDRS protocol and the result section does not report percent correct. I recommend the authors explicitly describe the limitations of ETDRS results and clearly separate ETDRS results from the Landolt C VA results by placing them in a different table.

We explicitly state in the manuscript that prosthetic visual acuity was assessed using Landolt C optotypes. Paragraph 9 of the main text: “Prosthetic visual acuity with PRIMA-2 glasses was measured using Landolt C optotypes”. Methods, section Assessment of prosthetic vision: “Visual acuity was assessed using a computer-generated Landolt C in 4 different orientations (gap at the top, bottom, right or left), so that a random response corresponds to 25% accuracy. The threshold optotype size was defined as the proper symbol recognition with at least 62.5% accuracy”.

I suspect that confusion regarding the ETDRS arises from the Supplementary Video S1, which is just an illustration of the patient’s performance, as we mentioned in the previous response to review. To avoid any confusion, we renamed it now to “Illustration of the patient’s letter recognition” rather than “Letter acuity test”. If you prefer, we could even remove this video from the article altogether.

2. The last paragraph should include a section on limitations of the study. Limitations include
 - a. The lack of take home testing limits the cumulative active time for PRIMA to a few hundred hours. Thus, claims of safety and stability for 24 months need to be qualified with a statement that the PRIMA was not used chronically, but rather implanted chronically and used occasionally.

Yes, this feasibility study was limited primarily to the clinical evaluation of prosthetic vision. Future study will involve a larger number of patients and will be expanded to the home use. We added a statement about this study limitation (see below).

- b. The zoom test did not include a control condition for comparison. I agree with the authors’ rebuttal that an electronically magnified display is the correct control. Such devices are readily available with commercial systems such as NuEyes and e-sight, and the AR glasses used for PRIMA can likely support this function. Is PRIMA an improvement over an electronically magnified visible light display, which does not require an implant?

Electronic magnifying glasses are designed to project the image onto the residual peripheral retina, where resolution rapidly decreases with eccentricity. On the contrary, PRIMA implant is designed to reintroduce the central vision, in conjunction with the remaining peripheral vision. This feasibility study was designed to assess the prosthetic vision with the first version of the implant, but more studies will be required to evaluate all the practical functional aspects and potential benefits of the system, compared to various alternatives, especially with its home use. We added a statement about this study limitation (see below).

c. An improvement in functional vision is not demonstrated in these patients, considering the level of vision available in the fellow eye. Therefore, at its current state, PRIMA will have limited impact, since patients will likely use the natural vision in their better eye for tasks that require high acuity vision. The author note several on-going improvements to address this issue.

Yes, resolution of the first version of the implant is limited by its pixel size to no better than 20/420. Future implants with smaller pixels may improve the resolution further.

We now included a paragraph regarding these study limitations at the end of the manuscript – right before the Summary: “This study was limited primarily to the clinical evaluation of prosthetic vision. Future testing will be expanded to include a home use, which will increase the active time and also help assess the functional benefits of the device, considering the residual vision available in the fellow eye. Future studies will also assess a practical value of the electronic magnification for the central vision in the PRIMA system.”

Reviewers' Comments:

Reviewer #5:

Remarks to the Author:

I appreciate the limitations paragraph added to the manuscript.

The authors insist on treating anecdotal information (letter recognition) the same as rigorously collected data (Landolt C). The Landolt C data is well described and collected following a methodical procedure. The letter reading demonstration is not data, it is an anecdote. Yet, the authors want to include these anecdotes in a data table. This is misleading. Misleading claims like this must be avoided, but especially when reporting clinical evaluation of a commercial device. Inclusion of such information, including videos, is acceptable and in fact helpful, but it must be described appropriately. Using letters as a target is reasonable, since restoring reading is an important clinical outcome. But, the authors use these anecdotal observations to claim a level of visual acuity with PRIMA as well as an improvement in visual acuity due to PRIMA, but these claims are not supported by rigorously collected data. Therefore, these claims should not be in Table 1.

To be specific, my recommended changes are

1a. Explicitly state that the letter recognition capability was not systematically evaluated and describe the results as preliminary.

or

1b. Include more information on percentage correct when reading letters

2. If 1a is chosen, then remove the lines from Table 1 "PRIMA-2 (AR), Landolt VA with preferred magnification, 18-24 months"
and "LogMAR gain due to PRIMA 18-24 months"

We would like to thank the editors for accepting our manuscript in principle.

In response to the editors' request "please address the first suggestion of the reviewer by explicitly stating that the letter recognition capability was not systematically evaluated and describe the results as preliminary, and please don't present them in the main results table alongside the pre-specified more robust analyses", we now further clarified that the quantitative measurements of prosthetic acuity presented in this paper were achieved only with Landolt C optotypes and not with letter charts. This is explicitly stated in the main text of the article, in the Methods and in Table 1. To avoid any confusion, we now deleted all mention of the "letter" and replaced that word with "Landolt C optotype" in description of these results. Letter recognition test is illustrated only in the Supplemental Materials and not in the main Figures or Results. In addition, we now added further clarification after mentioning these supplementary videos: "We would like to emphasize that these Supplementary videos are for illustration purposes only. Quantitative measurements of the prosthetic acuity were conducted with Landolt C optotypes".

Moreover, in response to a similar comment by this reviewer in the previous round of review, we offered removing these Supplementary videos altogether, so there would be no illustration of the letter recognition at all. However, the reviewer responded that "Inclusion of such information, including videos, is acceptable and in fact helpful, but it must be described appropriately." So, we kept the Supplementary videos and now added an explicit disclaimer: "We would like to emphasize that these videos are for illustration purposes only. Quantitative measurements of the prosthetic acuity were conducted with Landolt C optotypes".